# Vulnerability Analysis of LTE-R Train-to-Ground Communication Time Synchronization

**Yong Chen \*, Zhixian Zhan and Kaiyu Niu**

School of Electronics and Information Engineering, Lanzhou Jiaotong University, Lanzhou 730070, China; amy102607@163.com (Z.Z.); nky924212625@163.com (K.N.)
* Correspondence: edukeylab@126.com

**Abstract:** The time synchronization of LTE-R train-to-ground communication systems plays an important role in ensuring the safety of high-speed railways. In the LTE-R time synchronization process, existing problems, such as the time synchronization message broadcast address and LTE-R all-IP architecture, are vulnerable to attack. In order to analyze the impact of these problems, we propose a new vulnerability analysis method of LTE-R time synchronization based on stochastic Petri nets. Firstly, we construct a stochastic Petri net model of an LTE-R time synchronization process under attack. Secondly, steady-state probability expressions are obtained using the model isomorphism Markov chain. Finally, bychanging the firing rate of several key vulnerable nodes, the relationship curve between the firing rate and the steady-state probability of the clock node is obtained. Simulations show that the vulnerability of LTE-R time synchronization is most affected by the attack on eNodeB of the LTE-R base station. The results can provide a certain theoretical basis for the evolution of high-speed railway GSM-R communication systems to LTE-R.

**Keywords:** high-speed railway; time synchronization; LTE-R; stochastic Petri nets; vulnerability analysis

## 1. Introduction

At present, train-to-ground communication systems of high-speed railways use the global system for mobile communications-railway (GSM-R). However, GSM-R has the problems of narrow bandwidth, less carrying services, and low throughput, making it difficult to meet the requirements for the development of intelligent and automated high-speed railways [1]. The International Union of Railways (UIC) pointed out that GSM-R should evolve to the long-term evolution for railway (LTE-R) [2]. LTE-R is the next generation of high-speed railway train-to-ground communication systems, and it adopts a flatter network architecture and all-IP packet switching modes. Compared with GSM-R, LTE-R significantly reduces network complexity and construction cost.

LTE-R carries a large number of traffic safety control commands. Realizing LTE-R time synchronization is the key to ensuring the safe operation of a high-speed railway [3]. At present, the GSM-R wireless communication system uses the network time protocol (NTP) to complete train-to-ground time synchronization [4]. However, the NTP has some disadvantages, such as a low synchronization accuracy, large delay, and poor stability of the transmission process, which will make it difficult to meet the high-precision time synchronization requirements of LTE-R systems in the future [5]. Because of the above problems, in fields with high requirements for time synchronization accuracy, the use of the precision time protocol (PTP) instead of the NTP has been studied [6,7]. Kong [8] applied PTP technology to the test section of China's Shuohuang railway. The test result showed that PTP time synchronization technology can meet the use requirements of the wireless packet network of China's Shuohuang railway.

However, the PTP adopts a multicast address and LTE-R all-IP architecture, which are more vulnerable to network attacks [9]. Han et al. [10] pointed out that denial of service

(DoS) attacks, delay attacks, modification attacks, and spoofing attacks can force the clock to align with the wrong time or directly interrupt the synchronization process. Yaoet al. [11] proposed that an attacker can establish a connection with the client by forging and spoofing the server, and then they can achieve the purpose of arbitrarily manipulating the client time. Yu et al. [12] pointed out that an attacker can force the device to actively refuse to serve the target LTE terminal by sending a large number of signaling messages so as to complete a DoS attack. Decusatis et al. [13] proposed that DoS attacks can be launched by sending spam to the target slave station through deceptive data packets. Narula et al. [14] stated that encrypting the PTP can prevent some attacks but cannot prevent delay attacks, man-in-the-middle (MITM) attacks, and other attacks.

In summary, atime synchronization process using the PTP will inevitably be subjected to various attacks. Therefore, in view of the above characteristics of the PTP, the major aim of this article is to analyze the vulnerable nodes that affect the LTE-R time synchronization process by using a combination of stochastic Petri nets (SPNs) and a continuous-time Markov chain (CTMC), andarelationship curve between the average firing rate of vulnerable nodes and the steady-state probability of each normal and abnormal end state of the synchronization process is obtained. This method is able to identify the most vulnerable nodes in the LTE-R time synchronization process.The specific steps of the method are as follows:Firstly, we establish an SPN model of anLTE-R time synchronization process under attack. Then, we use the method of model isomorphism to transform the SPN model into a Markov chain (MC). Last but not least, by analyzing the relationship between the average firing rate of several key vulnerable nodes and the steady-state probability of the time node, we determine the key factors that affect the vulnerability of the LTE-R time synchronization process. By using this method, we draw the conclusion that an attack on eNodeB of the LTE-R base station affects the time synchronization process the most. This result can provide a certain reference basis for the evolution of the GSM-R time synchronization network to LTE-R.

## 2. Network Architecture and Basic Knowledge

### 2.1. GSM-R and LTE-R Network Architectures

As the next generation of high-speed railway wireless communication, LTE-R has an adjusted network architecture compared with GSM-R, and it has adopted a flatter network structure. LTE-R combines the base station controller (BSC) and base transceiver station (BTS) in GSM-R into evolved Node B (eNodeB), making the networking architecture of LTE-R flat. GSM-R's and LTE-R's specific network architectures are shown in Figure 1.

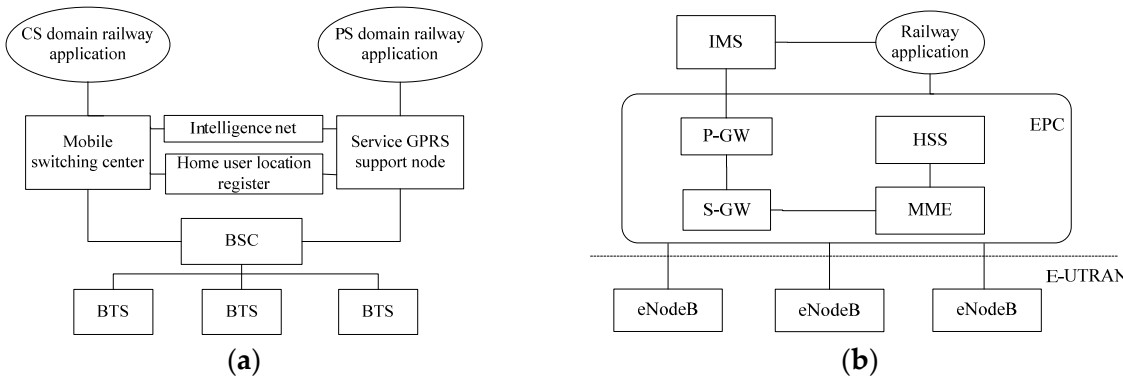

**Figure 1.** The network architecture of (**a**) GSM-R and (**b**) LTE-R.

Compared with GSM-R, LTE-R is a typical broadband system, which has a significantly improved transmission rate, transmission delay, service firing, and performance. In terms of transmission rate, GSM-R only has a transmission speed of 2400 bit/s to 9600 bit/s under the bandwidth of 200 kHz, while LTE-R has a downlink peak transmission rate of 100 Mbit/s and an uplink peak transmission rate of 50 Mbit/s under the bandwidth of 20 MHz [15]. In

terms of technology, LTE-R uses orthogonal frequency division multiplexing (OFDM) and multiple-input multiple-output (MIMO) technology, which not only solves the problem of serious frequency shortage but can also provide a high transmission rate.

### 2.2. PTP Time Synchronization Principle

PTP is proposed for distributed network measurements and control systems [16]. It adopts the technology of a physical layer hardware timestamp and is composed of a master clock sending PTP messages and a slave clock receiving the messages. The master and slave clocks determinethe clock deviation between them by exchanging PTP synchronization messages with a timestamp, and the slave clock compensates for this deviation. Finally, the time synchronization of the master–slave clock is completed. The interaction process of a PTP time synchronization message is shown in Figure 2.

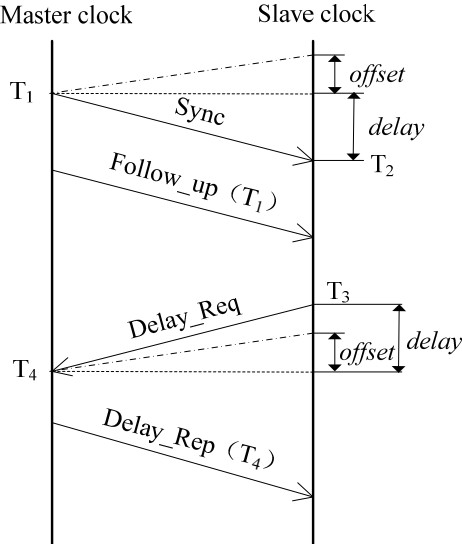

**Figure 2.** PTP time synchronization message interaction process.

In Figure 2, the specific mechanism of the PTP time synchronization process is as follows:

1. The master clock sends a Sync message to the slave clock in the form of a broadcast at $T_1$, and the slave clock receives the message at $T_2$;
2. Then, the master clock sends a Follow_up message to the slave clock in the form of a broadcast, which carries the sending $T_1$ timestamp of the Sync message;
3. The slave clock sends a Delay_Req message to the master clock in a point-to-point manner at $T_3$, and the master clock receives the message at $T_4$;
4. The master clock sends a Delay_Rep message to the slave clock in a point-to-point manner, which carries the $T_4$ timestamp of the master clock reception time.

From the above PTP time synchronization protocol interaction process, we obtain

$$T_2 = T_1 + offset + delay \tag{1}$$

$$T_4 = T_3 - offset + delay \tag{2}$$

where $T_i$ ($i$ = 1, 2, 3, 4) represents the timestamp of the master–slave clock receiving and sending PTP packets, *offset* represents the offset of the master and slave clocks, and *delay* represents the path delay of the packet transmission between the master and slave clocks.

Through Equations (1) and (2), the time *delay* and *offset* values can be calculated as follows:

$$delay = \frac{(T_2 - T_1) + (T_4 - T_3)}{2} \tag{3}$$

$$offset = \frac{(T_2 - T_1) - (T_4 - T_3)}{2} \tag{4}$$

According to the *delay* and *offset* values obtained from Equations (3) and (4), the size of the local clock is continuously adjusted to complete the time synchronization process.

*2.3. LTE-R Train-to-Ground Communication Time Synchronization Process*

The LTE-R train-to-ground communication time synchronization process adopts the master–slave response mode. eNodeB is used as the primary clock node, and the on-board controller (OBC) of the high-speed train is used as the secondary clock node. eNodeB communicates with the high-speed train OBC through the LTE-R wireless transmission channel and the vehicle station (VS) deployed on the roof. Thus, the time synchronization of train-to-ground communication is proposed as shown in Figure 3.

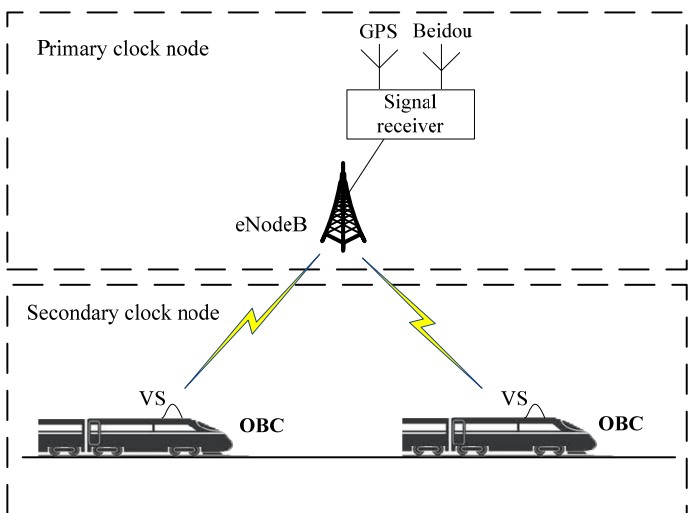

**Figure 3.** The time synchronization process of LTE-R train-to-ground communication.

In Figure 3, eNodeB is set as the primary clock node to obtain accurate time information by using GPS or Beidou satellite signals as the external clock source. The OBC is set as the secondary clock node. The master–slave clock establishes two-way communication through the all-IP network of the LTE-R wireless channel, and it transmits a PTP time synchronization message to achieve time synchronization between the eNodeB and OBC train-to-ground communication.

## 3. Attack Analysis of Time Synchronization Process of LTE-R Train-to-Ground Communication

The time synchronization process of LTE-R train-to-ground communication is vulnerable to attack. The main reasons for this are as follows:

- LTE-R has an all-IP architecture. The LTE-R railway wireless communication system is designed with an all-IP architecture, and the IP protocol is an unreliable packet communication protocol. It has a potential fault whereby the PTP message loss and wrong sequence caused by an attacker's intrusion are not detected. The attacker can continuously attack the PTP synchronization process, and this is not easily detected. This problem seriously affects the traffic safety of high-speed railways.
- The PTP sends synchronization messages in the form of multicast addresses [17]. Attackers can make full use of this feature to monitor and obtain the synchronization messages sent by the master clock. Moreover, attackers can make a spoofing packet with the frame format shown in Figure 4 to complete two-way deception between the master and slave clocks. This method is also imperceptible.

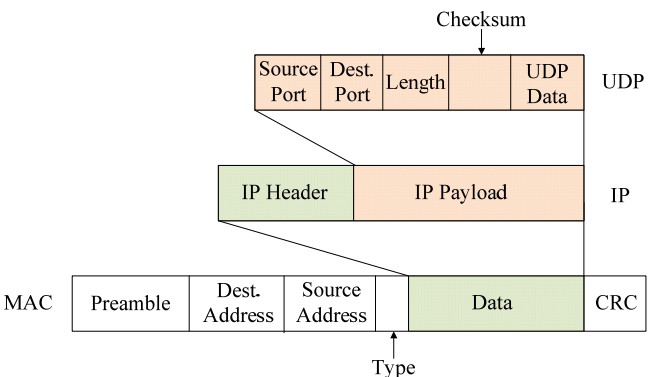

**Figure 4.** MAC frame format of PTP message.

In addition to meeting the frame structure depicted in Figure 4, the forged data packet also needs to meet the characteristics of the MAC destination address being a broadcast address and the message type being UDP. Due to the above factors, in the LTE-R time synchronization process, the attacker can imperceptibly join the PTP message interaction process between eNodeB and the OBC by forging the PTP message to complete the interception, maliciously tamper with the system, and delay the sending of normal PTP time synchronization messages. The time reference between eNodeB and the OBC drifts, resulting in inconsistent clocks of the train-to-ground equipment based on LTE-R, which seriously affects the safety of train operation.

## 4. Establishment of LTE-R Time Synchronization SPN and MC Models

### 4.1. Basic Concepts of Stochastic Petri Nets

An SPN is generally defined as a six-tuple array $SPN = (P, T, F, K, M, \lambda)$, where:

- $P = \{P_1, P_2, \ldots, P_n\}$ is a finite set of repositories, and $n$ is the number of repositories;
- $T = \{t_1, t_2, \ldots, t_n\}$ is a finite set of changes, and $m$ is the number of changes, satisfying $P \cap T = \Phi$ and $P \cup T = \Phi$;
- $F \subseteq I \cup O$ is a set of directed arcs, where $I$ represents the set of transition input arcs, $I \subseteq P \times T$; $O$ represents the set of transition output arcs, $O \subseteq T \times P$; the forbidden arc is allowed in $F$; and the forbidden arc only exists in the arc from the depot to the transition;
- $W:F \rightarrow N^+$ is an arc function, where $N^+(1, 2, 3, \ldots )$;
- $M:P \rightarrow N$ is the marking of the Petri net. $M_0$ is the initial marking, which indicates the initial state of the system;
- $\lambda = \{\lambda_1, \lambda_2, \ldots, \lambda_m\}$ represents the average firing rate set associated with the transition.

### 4.2. Modeling Analysis of LTE-R Time Synchronization Vulnerability Based on Stochastic Petri Nets

Due to the all-IP architecture of LTE-R and the broadcast sending address of the PTP, an ARP attack can easily be carried out at the data link layer during LTE-R time synchronization. The attacker becomes a middleman by forging the MAC frame structure and other means, imperceptibly inserting it into the normal communication process between eNodeB and the OBC, and maliciously tampering with or delaying the transmission of normal interactive messages randomly. This has a great impact on the synchronization accuracy of the LTE-R time synchronization process, and it seriously endangers the train operation safety and affects the real-time performance of train control systems. However, because it is difficult to find the attacker and the tampering of the message by the attacker is random and uncertain, it is difficult for the general method to accurately describe the attacker's behavior dynamically.

Stochastic Petri nets can build a complete LTE-R time synchronization process model under attack, restore the impact of the attack on the synchronization process, realize the

vulnerability analysis of the attacker on the LTE-R time synchronization process, and then determine the most vulnerable nodes in the synchronization process. Through the special protection of these key nodes, the success rate of attacks can be reduced, and train operation safety can be better guaranteed.

### 4.3. SPN Model of LTE-R Time Synchronization Process

The steps to establish an LTE-R time synchronization scheme model based on SPN are as follows:

1.  Establish the SPN model of the LTE-R train-to-ground communication time synchronization process according to Figures 2 and 3;
2.  Analyze the reachability set of the SPN model. Transform the actual transition marked on each arc into its average firing rate, and construct a continuous-time Markov chain;
3.  Solve the steady-state probability according to the related theorem of Markov chain stationary distribution and Chapman–Kolmogorov equations. Suppose the steady-state probability of $n$ reachable states is $P[M_i] = x_i (1 \leq i \leq n)$. Determine the element $x_i$ in the steady-state probability set $X = [x_1, x_2, \dots, x_n]$ using the following system of equations:

$$\begin{cases} XQ = 0 \\ \sum_{i=1}^{n} x_i = 1 \end{cases} \tag{5}$$

where $Q$ is the transfer rate matrix of the Markov process, and $n$ is the total number of states.

4.  Substitute $\lambda = \{\lambda_1, \lambda_2, \dots, \lambda_m\}$, solve the equations, solve the stability probability of each state, and analyze the LTE-R time synchronization process according to the obtained steady-state probabilities.

The LTE-R train-to-ground communication time synchronization process adopts a secondary clock node, in which eNodeB is the master clock and the OBC is the slave clock. They receive and forward time synchronization messages through the PTP time synchronization protocol. According to the operation mechanism of the LTE-R time synchronization PTP and the vulnerable characteristics of the PTP multicast address and all-IP architecture, a vulnerability analysis model of anLTE-R time synchronization process under attack based on the SPN theory is established. The SPN model is shown in Figure 5. The definitions of the SPN model places are shown in Table 1, and the definitions of transitions are shown in Table 2.

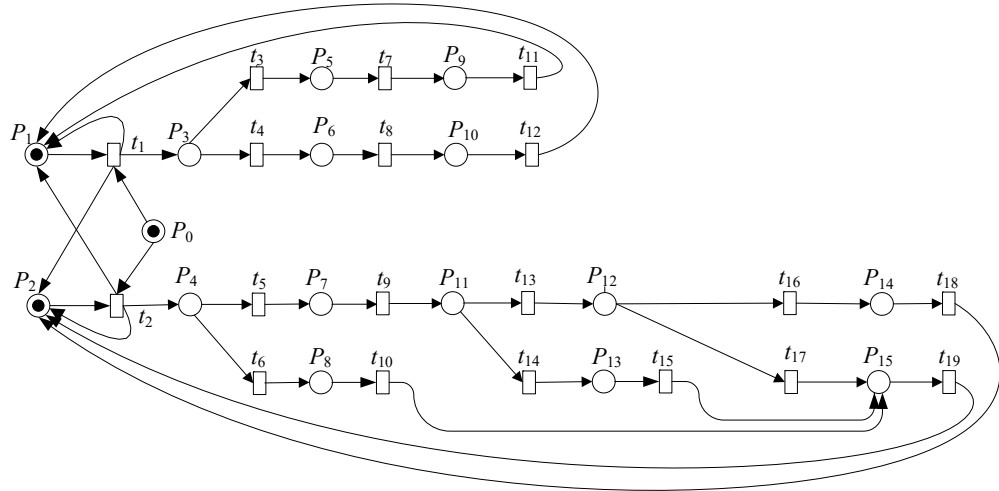

**Figure 5.** SPN model of LTE-R time synchronization process under attack.

**Table 1.** Definitions of places.

| Places | Definition |
|---|---|
| $P_0$ | Attacker intrusion status |
| $P_1$ | Initialization of the primary clock node eNodeB |
| $P_2$ | Initialization of the secondary clock node OBC |
| $P_3$ | eNodeB waits to receive the Delay_Req message |
| $P_4$ | OBC waits to receive Follow_up message |
| $P_5$ | eNodeB receives the malicious tampered PTP message |
| $P_6$ | eNodeB receives the Delay_Rep message |
| $P_7$ | Complete the process of calculating clock offset |
| $P_8, P_{13}$ | OBC receives the malicious tampered PTP message |
| $P_9$ | eNodeB time synchronization failed, entering the next cycle |
| $P_{10}$ | eNodeB time synchronization succeeded, entering the next cycle |
| $P_{11}$ | OBC waits to receive the Delay_Req message |
| $P_{12}$ | Complete the process of calculating clock delay |
| $P_{14}$ | OBC time synchronization succeeded, entering the next cycle |
| $P_{15}$ | OBC time synchronization failed, entering the next cycle |

**Table 2.** Definitions of transitions.

| Transitions | Definition |
|---|---|
| $t_1, t_2$ | The primary and secondary clock nodes eNodeB and OBC establish communication connections with each other |
| $t_3, t_6, t_{14}$ | Clock nodes receive the malicious tampered PTP synchronization message |
| $t_4$ | eNodeB receives Delay_Req message |
| $t_5$ | OBC receives Follow_up message and calculates the value of *offset* |
| $t_7, t_{10}, t_{15}$ | Exception handling of tampered message |
| $t_8$ | eNodeB sends Delay_Rep message |
| $t_9$ | OBC sends Delay_Req message |
| $t_{11}$ | eNodeB abnormal end |
| $t_{12}$ | eNodeB normal end |
| $t_{13}$ | OBC receives Delay_Rep message and calculates the value of *delay* |
| $t_{16}$ | It is determined that *offset* and *delay* meet the time threshold |
| $t_{17}$ | It is determined that *offset* and *delay* do not meet the time threshold |
| $t_{18}$ | OBC normal end |
| $t_{19}$ | OBC abnormal end |

In this model, eNodeB and the OBC establish the uplink synchronization relationship between them by performing random access processes at $t_1$ and $t_2$, respectively. According to Section 3, the attacker $P_0$ can complete the ARP attack by forging the MAC frame structure, setting the destination address to the broadcast address, and modifying the message type to UDP. In this process, as a middleman, the attacker can insert themselves into the regular communication between eNodeB and the OBC and intercept the PTP message with a normal interaction timestamp. For the intercepted PTP message, the attacker can maliciously tamper with the timestamp information in the UDP data field in the MAC frame or delay the sending of the message.

In one synchronization cycle of the SPN model, the master clock node eNodeB waits to receive the Delay_Reqmessage $P_3$ sent by the OBC. If eNodeB receives a message without timestamp information in the UDP data field of the MAC frame, it is determined that the message has been maliciously tampered with by the attacker. The synchronization cycle fails, and the token reaches the $P_5$ state through transition $t_3$ and finally reachese NodeB synchronization failure state $P_9$. The synchronization cycle is completed and enters the next cycle. If a PTP message with timestamp information is received, it is considered that it has been sent by the OBC and has not been maliciously tampered with; then, the token at $P_3$ reaches the $P_6$ state through transition $t_4$, and then eNodeB sends theDelay_Rep message to the OBC at $t_8$. Finally, the token reaches the OBC completion time synchronization status $P_{10}$ and enters the next cycle.

In the same cycle, the slave clock node OBC waits to receive the synchronization cycle message $P_4$ sent by eNodeB. Similarly, if the received message has no timestamp information in the UDP data field of its MAC frame, it is determined to be a message that has been maliciously tampered with by the attacker. The OBC synchronization fails in this cycle, and the token reaches the $P_8$ state through $t_6$, then reaches the OBC time synchronization end failure state $P_{15}$, and enters the next cycle. If there is timestamp information in the UDP data field of the received message, it is determined that the OBC has received the normal Follow_up message sent by eNodeB. Then, the token arrives at $t_5$, OBC calculates the *offset* value according to the $T_1$ and $T_2$ timestamp information in the Follow_up message, and the token reaches the $P_7$ state. Similarly, the OBC waits to receive the delayed response message $P_{11}$ sent by eNodeB, and if there is no timestamp information in the UDP data of the received message, it is determined that the message has been maliciously tampered with; the OBC synchronization cycle fails, and the token reaches the $P_{13}$ state through $t_{14}$. Finally, the token reaches the synchronization failure state $P_{15}$ and enters the next cycle. If a message containing timestamp information is received, it is determined to be a normal message sent by eNodeB. The token enters $t_{13}$ to calculate the value of the path delay according to the received timestamp. Then, according to the synchronization deviation threshold, it is judged whether the offset and delay values are legal. If they are legal, the token enters $t_{16}$, reaches the OBC time synchronization completion state $P_{14}$, and enters the next cycle. If they are illegal, this indicates that the attacker launched a delay attack in the synchronization process, causing the synchronization process to fail, and the token reaches $P_{15}$ and enters the next cycle.

*4.4. Isomorphic MC Based on SPN*

When the SPN isomorphism is a continuous-time Markov chain, the isomorphism transformation is carried out according to the following steps:

1. Firstly, the firing rule between states in the SPN model is analyzed, and the marked reachable sets of all states are obtained;
2. Each marking of the SPN is converted into a node corresponding to the continuous-time Markov chain reachability graph;
3. The transition events between different markings in the SPN model are mapped into arcs between nodes of the CTMC reachability graph, showing the logical relationship between the different states of the system;
4. Then, the transition firing rate in the SPN model is marked on each arc in the reachability graph of CTMC, and the CTMC distribution probability function is obtained.

Firstly, the SPN model reachable sets are constructed. In the SPN model displayed in Figure 5, when the LTE-R time synchronization process does not undergo an ARP attack and is in the normal operation state, there is a token in $P_0$, $P_1$, and $P_2$ in the model, and the initial state $M_1$ can be marked as $M_1 = (0, 1, 2)$. When the attacker launches an ARP attack, the tokens at $P_0$, $P_1$, and $P_2$ begin to move. According to the relationship between the different transition events of the SPN model in Figure 5, the following state reachable sets can be obtained: $M_2 = (1, 2, 3)$, $M_3 = (1, 2, 5)$, $M_4 = (1, 2, 6)$, $M_5 = (1, 2, 9)$, $M_6 = (1, 2, 10)$, $M_7 = (1, 2, 4)$, $M_8 = (1, 2, 7)$, $M_9 = (1, 2, 8)$, $M_{10} = (1, 2, 11)$, $M_{11} = (1, 2, 13)$, $M_{12} = (1, 2, 12)$, $M_{13} = (1, 2, 15)$, and $M_{14} = (1, 2, 14)$. $M_1$–$M_{14}$ are 14 states of the LTE-R time synchronization SPN model, including the eNodeB abnormal end state $M_5$, eNodeB normal end state $M_6$, OBC abnormal end state $M_{13}$, and OBC normal end state $M_{14}$. According to the SPN model, they are transformed into the nodes of the CTMC reachability graph.

Then, the transition events $t_i$ converted between the different markers in the SPN model are mapped to the arcs between the nodes of the CTMC reachable graph, and the firing rates $\lambda_i$ are correspondingly marked on each arc in the CTMC reachable graph. $\lambda_1$–$\lambda_{19}$ are the average firing rates between the different states in the SPN model. The average firing rate of the different states $\lambda_i$ ($i = 1, 2, \ldots, 19$) depends on the occurrence of different changes. If a directed arc is used to represent the transformation of different marks or states, an MC equivalent to the SPN model can be obtained, as shown in Figure 6.

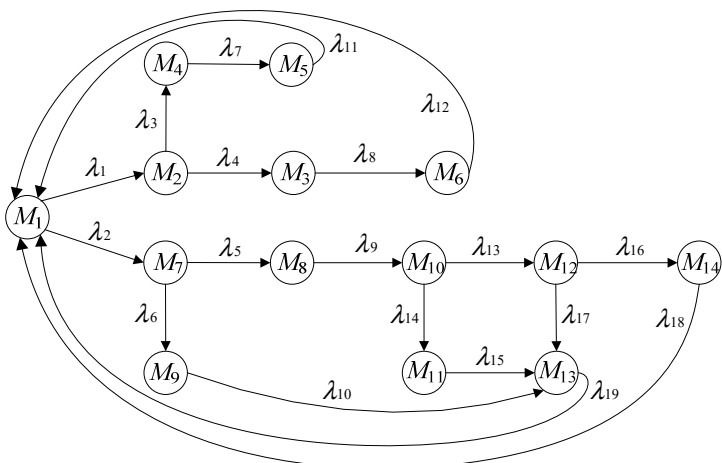

**Figure 6.** MC is isomorphic with the SPN model.

## 5. Vulnerability Analysis of LTE-R Time Synchronization

### 5.1. Calculate the Steady-State Probability of Each State of MC

$P(M_i)$ ($i$ = 1, 2, . . . , 14) is set as the steady-state probability of each state, $\lambda_i$ ($i$ = 1, 2, . . . , 19) is set as the average firing rate of different states, Equation (6) is obtained according to Equation (5), and Equation (6) is solved to obtain the steady-state probability of the LTE-R train-to-ground communication time synchronization process under attack. According to the values of these stability probabilities, the factors affecting the time synchronization process are further analyzed to analyze the vulnerability of the time synchronization process.

$$
\begin{cases}
\lambda_{11}P(M_5) + \lambda_{12}P(M_6) + \lambda_{18}P(M_{14}) + \lambda_{19}P(M_{13}) \\
\qquad = \lambda_1 P(M_1) + \lambda_2 P(M_1) \\
\lambda_1 P(M_1) = \lambda_3 P(M_2) + \lambda_4 P(M_2) \\
\lambda_4 P(M_2) = \lambda_8 P(M_3) \\
\lambda_3 P(M_2) = \lambda_7 P(M_4) \\
\lambda_7 P(M_4) = \lambda_{11} P(M_5) \\
\lambda_8 P(M_3) = \lambda_{12} P(M_6) \\
\lambda_2 P(M_1) = \lambda_5 P(M_7) + \lambda_6 P(M_7) \\
\lambda_5 P(M_7) = \lambda_9 P(M_8) \\
\lambda_6 P(M_7) = \lambda_{10} P(M_9) \\
\lambda_9 P(M_8) = \lambda_{13} P(M_{10}) + \lambda_{14} P(M_{10}) \\
\lambda_{14} P(M_{10}) = \lambda_{15} P(M_{11}) \\
\lambda_{13} P(M_{10}) = \lambda_{16} P(M_{12}) + \lambda_{17} P(M_{12}) \\
\lambda_{15} P(M_{11}) + \lambda_{10} P(M_9) = \lambda_{19} P(M_{13}) \\
\lambda_{16} P(M_{12}) = \lambda_{18} P(M_{14}) \\
\sum_{i=1}^{14} P(M_i) = 1
\end{cases}
\tag{6}
$$

### 5.2. Vulnerability Analysis

During LTE-R time synchronization, a large number of malicious attacks occur during the message interaction process between the master and slave clocks, especially when waiting to receive critical time synchronization messages [18]. Based on this, this paper takes all the time spent waiting to receive time synchronization messages in the time synchronization process as the vulnerable nodes. These nodes can affect the vulnerability of the LTE-R time synchronization process, as shown in Table 3.

**Table 3.** Definition of vulnerable nodes.

| Transitions | Corresponding Firing Rate | Definition |
|---|---|---|
| $t_4$ | $\lambda_4$ | eNodeB receives the Delay_Req message sent by the OBC |
| $t_5$ | $\lambda_5$ | OBC receives the Follow_up message sent by eNodeB |
| $t_{13}$ | $\lambda_{13}$ | OBC receives the Delay_Rep message sent by eNodeB |

In Table 3, $\lambda_4$, $\lambda_5$, and $\lambda_{13}$ are used as the average firing rates of receiving key PTP messages, and their valuesare determined by the attacker who successfully intercepted the message. By changing the average firing rate value $\lambda_i$, the operation of the attacker can be simulated and the relationship with the steady-state probability $P(M_i)$ value of the protocol exception and normal end state can be established through Equation (6).Through simulation experiments, we find that when the average firing rate $\lambda_i$ is greater than 30, the steady-state probability $P(M_i)$ value of each end state remains unchanged, which means that the $P(M_i)$ value of each state tends to be stable when $\lambda_i$ is 30. Therefore, we choose to simulate the operation of the attacker by changing the average firing rate $\lambda_i$ in the range of 0–30. Moreover, we establish the relationship between the different attack behaviors of the attacker and the steady-state probabilities $P(M_i)$ of the abnormal and normal states of the protocol through Equation (6).

The $P(M_i)$ value of the abnormal and normal end states of the clock nodes is set as the index to measure the vulnerability of the synchronization process. At that time, the smaller the $P(M_i)$ value of the abnormal end state and the larger the $P(M_i)$ value of the normal end state, the stronger the survivability of the clock node in the attack state and the more robust the time synchronization network protocol and vice versa. This index is used to study the impact of the firing rate change of each vulnerable node on the vulnerability of the LTE-R time synchronization network protocol.

## 6. Simulation Analysis

### 6.1. eNodeB Vulnerability Analysis

Among the three vulnerable nodes, , as eNodeB, receives the average firing rate of the PTP messages sent by the attackers. We take the value of as 0–30 and the other value of as 1, and solve them in turn. The specific calculation results are shown in Table 4. In order to see the impact of the different protocol vulnerabilities more intuitively, the data presented in Table 4are presented in a schematic diagram as shown in Figure 7.

**Table 4.** Changes in the steady-state probability of each end state when $\lambda_4$ changes.

| Steady-State Probability | Variation Range | Changing Amplitude | Change Trend |
|---|---|---|---|
| $P(M_5)$ | 0.1429–0.0064 | −0.1365 | Reduces Sharply |
| $P(M_6)$ | 0–0.1911 | 0.1911 | Rises Sharply |
| $P(M_{13})$ | 0.1071–0.1481 | 0.041 | Rises Slightly |
| $P(M_{14})$ | 0.0179–0.0247 | 0.0068 | Almost Unchanged |

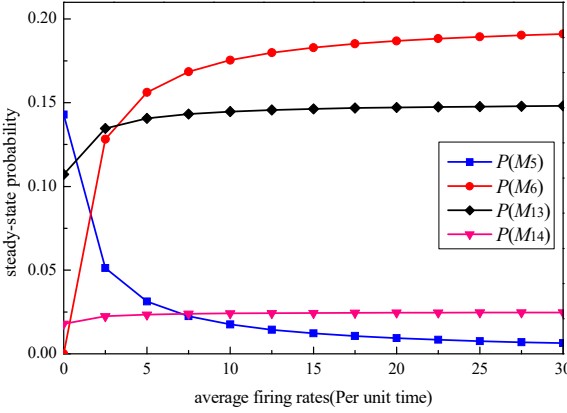

**Figure 7.** Simulation results of each steady-state probability when $\lambda_4$ changes.

As shown in Table 4 and Figure 7, the effects of different values on the steady-state probability values of each end state of the clock node are compared. As gradually increases, the steady-state probability values of each end state of eNodeB and the OBC change.Moreover, the steady-state probabilities $P(M_5)$ and $P(M_6)$ of the abnormal and normal end states of eNodeB change the most drastically. $P(M_5)$ drops rapidly by 0.1365,that is, from 0.1429 to 0.0064, and $P(M_6)$ increases rapidly by 0.1911 from the initial value of 0. Compared with the change amplitude in eNodeB, the steady-state probabilities $P(M_{13})$ and $P(M_{14})$ of the abnormal and normal end states of the OBC barely change, only by 0.041 and 0.0068, which are 1/3 and 1/28 of $P(M_5)$ and $P(M_6)$, respectively. This shows that the malicious tampering of the message by the attacker has the greatest impact on the device eNodeB of the message receiver.

### 6.2. OBC Vulnerability Analysis

Among the three vulnerable nodes, and are used as the OBC to receive the average firing rate of the PTP messages sent by the attackers. The values of and are taken as 0 and 30, respectively, and the other values are taken as 1. The specific calculation results are shown in Table 5. In order to see the impact of the different protocol vulnerabilities more intuitively, the data in Table 5 are presented in a schematic diagram as shown in Figure 8.

**Table 5.** Changes in the steady-state probability of each end state when and changes.

| Steady-State Probability | Fragile Node | Variation Range | Changing Amplitude | Change Trend |
|---|---|---|---|---|
| $P(M_5)$ | $\lambda_5$ | 0.0769–0.0769 | 0 | Unchanged |
| | $\lambda_{13}$ | 0.0714–0.0829 | 0.0115 | Almost Unchanged |
| $P(M_6)$ | $\lambda_5$ | 0.0769–0.0769 | 0 | Unchanged |
| | $\lambda_{13}$ | 0.0714–0.0829 | 0.0115 | Almost Unchanged |
| $P(M_{13})$ | $\lambda_5$ | 0.1538–0.0794 | −0.0744 | Reduces Slightly |
| | $\lambda_{13}$ | 0.1429–0.0856 | −0.0573 | Reduces Slightly |
| $P(M_{14})$ | $\lambda_5$ | 0–0.0372 | 0.0372 | Rises Slightly |
| | $\lambda_{13}$ | 0–0.0401 | 0.0401 | Rises Slightly |

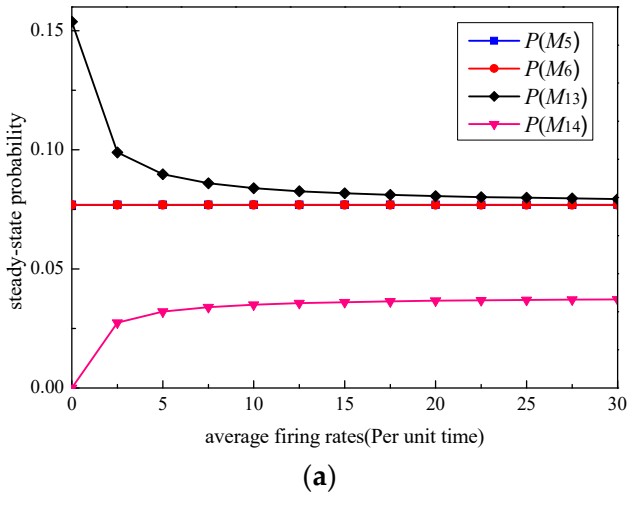 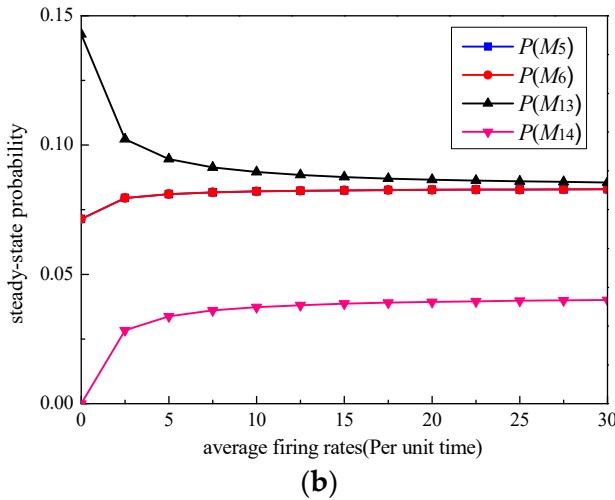

**Figure 8.** Simulation results of each steady-state probability when (**a**) changes and (**b**) changes.

As shown in Table 5 and Figure 8, the effects of different $\lambda_5$ and $\lambda_{13}$ values on the steady-state probability values of each end state of a clock node are compared. As the $\lambda_5$ value gradually increase, the steady-state probabilities $P(M_5)$ and $P(M_6)$ of the abnormal and normal end states of eNodeB remain unchanged; the steady-state probability $P(M_{13})$ of the abnormal end state of the OBC slightly decreases from 0.1538 to 0.0744, and the steady-state probability $P(M_{14})$ of the normal end state of the OBC slightly increases from 0 to 0.0372. Furthermore, as the $\lambda_{13}$ value gradually increase, the steady-state probability

$P(M_5)$ and $P(M_6)$ of the abnormal and normal end states of eNodeB slightly increase by 0.0115, the steady-state probability $P(M_{13})$ of the abnormal end states of the OBC decrease by 0.0573 from 0.01429, and the steady-state probability $P(M_{14})$ of the normal end states of the OBC slightly increase by 0.0401 from the initial value of 0. By comparing the changes in amplitude of the steady-state probability of the abnormal and normal end states of eNodeB and the OBC, it can be seen that the changes in $\lambda_5$ and $\lambda_{13}$ values have little impact on the steady-state probability of each end state of eNodeB, which shows that the OBC, as the slave clock of eNodeB, receives PTP messages with different degrees of attack and has little impact on the master clock of eNodeB. This conclusion is consistent with the conclusion of a previous study [13], where an experimental test platform was built to carry out ARP spoofing attacks on PTP synchronization messages. It is proved that the vulnerability analysis model of the LTE-R time synchronization network protocol in this paper is effective.

### 7. Conclusions

LTE-R is the next generation of high-speed railway wireless communication systems. However, due to the all-IP architecture of LTE-R and the use of a multicast address in the PTP, the time synchronization process of LTE-R communication is very vulnerable to attack. In this paper, an LTE-R train-to-ground communication vulnerability analysis method based on the combination of stochastic Petri nets and a continuous-time Markov chain is proposed, an analysis model of LTE-R time synchronization vulnerability based on stochastic Petri nets is established, and the Markov chain is obtained using the method of model isomorphism. The relationship between the average firing rate in the attack state and the steady-state probability of the end state of the clock node is quantitatively obtained, and the following conclusions are drawn:

1.  The master–slave clock is a vulnerable node in the whole process when it is waiting to receive the key PTP message. If it is attacked to varying degrees at this critical time, the whole synchronization process will be greatly impacted.
2.  When the master clock of the synchronization process, eNodeB, is attacked to varying degrees, the end states of the whole synchronization process are affected. However, when the slave clock, the OBC, whose timing structure is lower than that of eNodeB, is attacked to varying degrees, only the OBC is affected. This conclusion is consistent with the conclusion drawn in a previous study that conducted a physical experiment, further verifying the effectiveness of the SPN analysis model in this paper.

**Author Contributions:** Conceptualization, Y.C. and Z.Z.; methodology, Y.C. and Z.Z.; writing—review and editing, Z.Z. and K.N.; visualization, Z.Z. and K.N.; supervision, Y.C.; project administration, Z.Z.; funding acquisition, Y.C. All authors have read and agreed to the published version of the manuscript.

**Funding:** This research was supported by the National Natural Science Foundation of China (Grant No. 61963023 and No. 61841303) and the Tianyou innovation team of Lanzhou Jiaotong University (Grant No. TY202003).

**Institutional Review Board Statement:** Not applicable.

**Informed Consent Statement:** Informed consent was obtained from all subjects involved in the study.

**Data Availability Statement:** The study did not report any data.

**Conflicts of Interest:** The authors declare no conflict of interest.

## Abbreviations

The following abbreviations are used in this manuscript:

| | |
|---|---|
| ARP | Address Resolution Protocol |
| BSC | Base station controller |
| BTS | Base transceiver station |
| CTMC | Continuous-time Markov chain |
| DoS | Denial of service |
| eNodeB | evolved Node B |
| EPC | Evolved Packet Core |
| E-UTRAN | Evolved Universal Telecommunication Radio Access Network |
| GPRS | General Packet Radio Service |
| HSS | Home Subscriber Server |
| IMS | Information Management System |
| LTE-R | Long-term evolution for railway |
| MC | Markov chain |
| MME | Mobility Management Entity |
| MIMO | Multiple-input multiple-output |
| MITM | Man-in-the-middle |
| NTP | Network time protocol |
| OBC | On-board controller |
| OFDM | Orthogonal frequency division multiplexing |
| P-GW | PDN GateWay |
| PTP | Precision time protocol |
| S-GW | Serving GateWay |
| SPN | Stochastic Petri net |
| VS | Vehicle station |

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
