# Peer review of "Vulnerability Analysis of LTE-R Train-to-Ground Communication Time Synchronization"

_applsci, doi:10.3390/app12115572_

Round 1

Reviewer 1 Report

In this paper, an LTE-R train to ground communication vulnerability analysis method based on the combination of stochastic Petri net and continuous-time Markov chain is proposed, an analysis model of LTE-R time synchronization vulnerability based on stochastic Petri net is established, and the Markov chain is obtained by the method of model isomorphism. The conclusions could be supported by the results.

Author Response

Reviewer#1: In this paper, an LTE-R train to ground communication vulnerability analysis method based on the combination of stochastic Petri net and continuous-time Markov chain is proposed, an analysis model of LTE-R time synchronization vulnerability based on stochastic Petri net is established, and the Markov chain is obtained by the method of model isomorphism. The conclusions could be supported by the results.

Author response: Thank you very much for taking the time out of your busy schedule to review this article and for acknowledging it.

Author action: We did not modify the manuscript this issue.

Reviewer 2 Report

Title: Vulnerability analysis of LTE-R train to ground communication 2
time synchronization

  • Figures are not clear, please increase the quality of the figures
  • Use proper punctuations
  • Highlight the novelty
  • Define each parameter in all equation
  • Improve the grammar
  • How did you find the variation ranges?
  • Provide more details of the Tables

Author Response

Reviewer#2, Concern # 1: Figures are not clear, please increase the quality of the figures.

Author response: Thank you very much for your valuable comments. In the original manuscript, we ignored the size of some figures, which made them look unclear.

Author action: We update the manuscript by modifying some figures in the article that are not appropriate in size. The modified figures mainly include: Figure 2, Figure 4, Figure 7 and Figure 8.

Reviewer#2, Concern # 2: Use proper punctuations.

Author response: In the original manuscript, we didn't carefully check the punctuation in the whole article, resulting in the wrong use of some punctuations.

Author action: We updated the manuscript by modifying sentences with punctuation errors. The modified sentences are as follows:

Original sentence: Where Q is the transfer rate matrix of the Markov process; n is the total number of states;

Updated sentence: Where Q is the transfer rate matrix of the Markov process, and n is the total number of states.

Original sentence: OBC calculates the offset according to the T1 and T2 timestamp information in the Follow_up message and reaches the P7 state.

Updated sentence: OBC calculates the offset according to the T1 and T2 timestamp information in the Follow_up message, and reaches the P7 state.

Original sentence: By changing the average firing rate value , simulate the operation of the attacker, and establish the relationship with the steady-state probability P(Mi) value of protocol exception and normal end state through equation (6).

Updated sentence: We choose to simulate the operation of the attacker by changing the average firing rate  in the range of 0 to 30 and establish the relationship between the different attack behaviors of the attacker and the steady-state probability P(Mi) of the abnormal and normal states of the protocol through the equation (6).

Reviewer#2, Concern # 3: Highlight the novelty.

Author response: In the original document, we did not clearly describe in detail the novel contributions of this article.

Author action:We updated the manuscript by adding a description of the main contributions of this article. The added contents is as follows:

Therefore, in view of the above characteristics of the PTP protocol, the major contribution of this article is to analyze the vulnerable nodes that affect the LTE-R time synchronization process by using a combination of stochastic Petri nets (SPN) and continuous-time Markov chain (CTMC), and the relationship curve between the average firing rate of vulnerable nodes and the steady-state probability of each normal and abnormal end state of the synchronization process is obtained. This method is able to identify the most vulnerable nodes in the LTE-R time synchronization process.

Reviewer#2, Concern # 4: Define each parameter in all equation.

Author response: In the original manuscript, we did not fully explain the variables involved in each equation.

Author action: We updated the manuscript by adding the description of the variables in each equation. The added contents are as follows:

For equations 1 and 2:

Added contents: Where Ti (i = 1, 2, 3, 4) represents the timestamp of the master-slave clock receiving and sending PTP packets, offset represents the offset of the master and slave clocks, delay represents the path delay of packet transmission between master and slave clocks.

For equations 3 and 4:

The variables are consistent with equations 1 and 2, so there is no content to add.

For equations 5:

Added contents: Suppose the steady-state probability of n reachable states is P[Mi] = xi( ). The element xi in the steady-state probability set X = [x1, x2, … , xn] can be determined by the following system of equations.

For equations 6:

Added contents: Set P(Mi) (i=1, 2, …, 14) as the steady-state probability of each state and λi (i=1, 2, …, 19) as the average firing rate of different states, the equations (6) can be obtained according to equation (5).

Reviewer#2, Concern # 5: Improve the grammar.

Author response: In the original manuscript, we didn't carefully check the grammar in the whole article, resulting in the wrong use of grammar.

Author action: We updated the manuscript by modifying some sentences with grammatical errors. The modified sentences are as follows:

Original sentence: To sum up, in the process of time synchronization using PTP protocol, it will inevitably be subjected to various attacks.

Updated sentence: To sum up, in the process of time synchronization using the PTP protocol, it will inevitably be subjected to various attacks.

Original sentence: By using this method, we can get the conclusion that the attack on the eNodeB of LTE-R base station can most affect the time synchronization process.

Updated sentence: By using this method, we can get the conclusion that the attack on the eNodeB of the LTE-R base station can most affect the time synchronization process.

Original sentence: The specific network architecture of GSM-R and LTE-R, as shown in Figure 1.

Updated sentence: GSM-R and LTE-R’s specific network architecture is shown in Figure 1.

Original sentence: It adopted the technology of physical layer hardware timestamp and is composed of the master clock sending PTP message and the slave clock receiving the message.

Updated sentence: It adopted the technology of physical layer hardware timestamp and is composed of the master clock sending the PTP message and the slave clock receiving the message.

Original sentence: The attacker can continuously affect the PTP synchronization process and is not easy to be find.

Updated sentence: The attacker can continuously affect the PTP synchronization process and is not easy to find.

Original sentence: And the definition of transitions are shown in Table 2.

Updated sentence: And the definition of transitions is shown in Table 2.

Original sentence: It is considered that the message has maliciously tampered.

Updated sentence: It is considered that the message has been maliciously tampered with.

Original sentence: The transition events ti converted between different markers in the SPN model is mapped to the arcs between the nodes of the CTMC reachable graph.

Updated sentence: The transition events ti converted between different markers in the SPN model are mapped to the arcs between the nodes of the CTMC reachable graph.

Original sentence: MC isomorphic with SPN model.

Updated sentence: MC is isomorphic with the SPN model.

Original sentence: The P(Mi) value of the abnormal and normal end state of the clock nodes is set as the index to measure the vulnerability of synchronization process.

Updated sentence: The P(Mi) value of the abnormal and normal end state of the clock nodes is set as the index to measure the vulnerability of the synchronization process.

Original sentence: Due to the all-IP architecture of LTE-R and the use of the multicast address in PTP protocol,…

Updated sentence: Due to the all-IP architecture of LTE-R and the use of the multicast address in the PTP protocol,…

Reviewer#2, Concern # 6: How did you find the variation ranges?

Author response: In the original manuscript, we didn't explain the value of the variation ranges in detail.

Author action: We updated the manuscript by adding a detailed explanation on the selection of variation ranges. The added context is as follows:

Through simulation experiments, we find that when the average firing rate  is greater than 30, the steady-state probability P(Mi) value of each end state remains unchanged, which means that the P(Mi) value of each state tends to be stable when  is 30. Therefore, we choose to simulate the operation of the attacker by changing the average firing rate  in the range of 0 to 30. And we establish the relationship between the different attack behaviors of the attacker and the steady-state probability P(Mi) of the abnormal and normal states of the protocol through the equation (6).

Reviewer#2, Concern # 7: Provide more details of the Tables.

Author response: In the original manuscript, we didn't describe the Tables in detail.

Author action: We updated the manuscript by adding a detailed description of some Tables. The added descriptions are as follows:

For Table 7:

As  gradually increases, the steady-state probability values of each end state of eNodeB and OBC change. And the steady-state probabilities P(M5) and P(M6) of the abnormal and normal end states of the eNodeB change most drastically. P(M5) dropped rapidly by 0.1365 from 0.1429 to 0.0064, and P(M6) increased rapidly by 0.1911 from the initial 0. Compared with the change amplitude of eNodeB, the steady-state probabilities P(M13) and P(M14) of abnormal and normal end states of OBC have little change, only 0.041 and 0.0068, which are 1/3 and 1/28 of P(M5) and P(M6) respectively.

For Table 8:

As  gradually increases, the steady-state probabilities P(M5) and P(M6) of the abnormal and normal end states of eNodeB remain unchanged, and the steady-state probability P(M13) of the abnormal end state of OBC slightly decreases from 0.1538 to 0.0744, the steady-state probability P(M14) of the normal end state of OBC slightly increases from 0 by 0.0372. As  gradually increases, the steady-state probability P(M5) and P(M6) of abnormal and normal end states of eNodeB slightly increased by 0.0115, and the steady-state probability P(M13) of abnormal end states of OBC decreased by 0.0573 from 0.01429, and the steady-state probability P(M14) of normal end states of OBC slightly increased by 0.0401 from the initial 0. By comparing the change amplitude of the steady-state probability of abnormal and normal end states of eNodeB and OBC, it can be found that the changes of  and  values have little impact on the steady-state probability of each end state of eNodeB.

Round 2

Reviewer 2 Report

The authors improve the paper as suggested, hence the paper can be accepted in its present form